# Cognitive Sequelae and Hippocampal Dysfunction in Chronic Kidney Disease following 5/6 Nephrectomy

**DOI:** 10.3390/brainsci12070905

**Published:** 2022-07-11

**Authors:** Yeon Hee Yu, Seong-Wook Kim, Hyuna Im, Se Won Oh, Nam-Jun Cho, Samel Park, Dae-Kyoon Park, Duk-Soo Kim, Hyo-Wook Gil

**Affiliations:** 1Department of Anatomy, College of Medicine, Soonchunhyang University, Cheonan-Si 31151, Korea; yuyeon0220@naver.com (Y.H.Y.); hyuna99012@sch.ac.kr (H.I.); mdeornfl@sch.ac.kr (D.-K.P.); 2Graduate School of New Drug Discovery & Development, Chungnam National University, Daejeon 34143, Korea; seongwook0205@gmail.com; 3Department of Radiology, Eunpyeong St. Mary’s Hospital, College of Medicine, The Catholic University of Korea, Seoul 03312, Korea; oasis1979@gmail.com; 4Department of Internal Medicine, Soonchunhyang University Cheonan Hospital, Cheonan 31151, Korea; chonj@schmc.ac.kr (N.-J.C.); samelpark17@schmc.ac.kr (S.P.)

**Keywords:** chronic kidney disease, hippocampus, neuronal plasticity, blood–brain barrier, sodium–hydrogen exchangers

## Abstract

Neurological disorders are prevalent in patients with chronic kidney disease (CKD). Vascular factors and uremic toxins are involved with cognitive impairment in CKD. In addition, vascular dementia-induced alterations in the structure and function of the hippocampus can lead to deficits in hippocampal synaptic plasticity and cognitive function. However, regardless of this clinical evidence, the pathophysiology of cognitive impairment in patients with CKD is not fully understood. We used male Sprague Dawley rats and performed 5/6 nephrectomy to observe the changes in behavior, field excitatory postsynaptic potential, and immunostaining of the hippocampus following CKD progression. We measured the hippocampus volume on magnetic resonance imaging scans in the controls (*n* = 34) and end-stage renal disease (ESRD) hemodialysis patients (*n* = 42). In four cognition-related behavior assays, including novel object recognition, Y-maze, Barnes maze, and classical contextual fear conditioning, we identified deficits in spatial working memory, learning and memory, and contextual memory, as well as the ability to distinguish familiar and new objects, in the rats with CKD. Immunohistochemical staining of Na^+^/H^+^ exchanger1 was increased in the hippocampus of the CKD rat models. We performed double immunofluorescent staining for aquaporin-4 and glial fibrillary acidic protein and then verified the high coexpression in the hippocampus of the CKD rat model. Furthermore, results from recoding of the field excitatory postsynaptic potential (fEPSP) in the hippocampus showed the reduced amplitude and slope of fEPSP in the CKD rats. ESRD patients with cognitive impairment showed a significant decrease in the hippocampus volume compared with ESRD patients without cognitive impairment or the controls. Our findings suggest that uremia resulting from decreased kidney function may cause the destruction of the blood–brain barrier and hippocampus-related cognitive impairment in CKD.

## 1. Introduction

Previous clinical studies have shown that psychiatric problems, including cognitive impairment and dementia, are prevalent in patients with chronic kidney disease (CKD) [1,2,3,4,5,6]. Notably, cognitive disorders in patients with CKD are causally related to cerebrovascular disease, anemia, secondary hyperparathyroidism, dialysis disequilibrium, and uremic toxins [4,7,8]. In a recent clinical study, brain–kidney crosstalk revealed the similarities of the vascular and hemodynamic system in the kidney and brain [9]. Many comorbidities, including hypertension, diabetes mellitus, and CKD, are risk factors for cognitive impairment. A recent study showed glycemic variability could influence brain perfusion, which causes a decrease in perfusion in the hippocampus, and may impact the rate of cognitive deterioration [10]. However, regardless of this clinical evidence, the pathophysiology of cognitive deficit in patients with CKD is not fully known.

The hippocampus in the brain is essential for learning and memory function. Hippocampal volume has been measured to verify the relevance of cognitive deficits in psychiatric diseases [11,12,13,14]. However, the presence of hippocampal volume loss has not been reported in human CKD. Behavioral and neurochemical studies in animals, including our previous findings, showed that the hippocampal area in the brain is vulnerable to the effects of uremia [15,16,17,18], suggesting that uremia may induce cognitive impairment. Synaptic plasticity in the hippocampus underlies memory formation and cognition. However, the effect of chronic CKD, usually induced by the long-term effects of uremia on hippocampal synaptic plasticity, is unknown.

Here, in order to verify whether CKD contributes to cognitive deficits, we firstly performed 5/6 nephrectomy (5/6 Nx) and established a stable rat model of CKD. We next measured cognitive dysfunction and intrinsic synaptic properties in the hippocampus by performing behavioral, anatomical, and electrophysiological tests in the CKD rat model. Interestingly, we observed deficits in novel object recognition (NOR) memory, spatial working memory, 24-h contextual fear memory, and hippocampal synaptic plasticity. Together, our findings clearly showed that uremia caused hippocampus-relevant cognitive impairment.

## 2. Materials and Methods

### 2.1. Experimental Animals

Male Sprague Dawley (SD) rats (8 weeks old) obtained from the Experimental Animal Center, Soonchunhyang University (Cheonan, South Korea), were used. All animals were provided with a commercial diet and water ad libitum under controlled temperature, humidity, and lighting conditions of 22 ± 2 °C, 55 ± 5%, and 12:12 light/dark cycle, respectively. All animal experiments were approved by the Administrative Panel on Laboratory Animal Care of Soonchunhyang University (permit no. SCH18-0056). All possible efforts were made to avoid the suffering of the rats and minimize the number used during the experiments. CKD was induced by 5/6 Nx, as described previously [18]. Briefly, the rats were placed in a chamber, and general anesthesia was delivered using 2.5% isoflurane in a mixture of 33% oxygen and 67% nitrous oxide. Adrenal glands were protected, and then the upper and lower poles of the left kidney were resected using surgical scissors. Right site unilateral nephrectomy was performed.

### 2.2. Serum Biochemical Assays

Blood samples for standard biochemical assay were collected at different time points (4 and 10 weeks (wks) after surgery) by cardiac puncture. All serum biochemicals were measured as described in the Appendix A. In our study, all groups obtained similar results to prior study results (data not shown) [18].

### 2.3. Behavioral Tasks

All rats were used for cognition-related behavioral tests at 4 wks and 10 wks after CKD induction. Behavioral tests were monitored and analyzed using the PC-based video behavior analysis and automated tracking software Noldus EthoVision 3.1.

#### 2.3.1. Novel Objective Recognition Test

Object exploration behavior for discrete novel objects was investigated as a form of exploration activity that is a basic characteristic of rodents. The detailed method is described in the Appendix A [19,20].

#### 2.3.2. Y-Maze with Special Cue

Spontaneous alternation was assessed in a Y-maze test, as described previously [21]. A hippocampus-dependent spatial working memory Y-maze was placed in an isolated room with an existing clue and a special cue above each arm. The detailed method is described in the Appendix A.

#### 2.3.3. Barnes Maze

The Barnes maze consisted of a circular platform (122 cm diameter) with 20 holes located at the perimeter at equal distances and was elevated 105 cm above the floor. The detailed method is described in the Appendix A [21,22].

#### 2.3.4. Classical Fear Conditioning Test

Contextual and cued fear conditioning and 24-h retrieval of contextual memory were performed [23,24]. The detailed method is described in the Appendix A

### 2.4. In Vivo fEPSP Recording

fEPSPs were recorded in the hippocampal CA1 region of the rats, as described in [25,26,27,28], with some modifications. The detailed method is described in the Appendix A.

### 2.5. Tissue Processing and Immunohistochemistry (IHC)

For the immunohistochemical experiments, animals were anesthetized (urethane 1.5 g/kg, i.p.) and perfused transcardially with phosphate-buffered saline (PBS) followed by 4% paraformaldehyde in 0.1 M PB. The brains were extracted and post-fixed in the same fixative for 4 h and rinsed in PB containing 30% sucrose at 4 °C for 2 days. Thereafter, the tissues were cryocutted using a microtome at 30 μm. The sections were incubated with the primary antibody in PBS containing 0.3 % Triton X-100 overnight at 4 °C. The primary antibodies were mouse antiglial fibrillary acidic protein (GFAP) IgG (Millipore, MA, USA; diluted 1:1000) and rabbit anti-Na^+^/H^+^ exchanger-1 (NHE-1) IgG (Millipore; diluted 1:200). The sections were washed three times for 10 min with PBS, incubated sequentially in biotinylated goat antirabbit IgG (Vector labs, CA, USA) or goat antimouse IgG (Vector) and ABC complex (Vector), and diluted 1:250 in the same solution as the primary antiserum. The sections were visualized with 3,3′-diaminobenzidine (DAB) in 0.1 M Tris buffer and mounted on gelatin-coated slides. The immunoreactions were observed using a DMRB microscope (Leica, Wetzlar, Germany), and images were captured using a model DP72 digital camera and DP2-BSW microscope digital camera software (Olympus, Tokyo, Japan) [29].

### 2.6. Double Immunofluorescent Staining

To identify the morphological changes in the blood–brain barrier (BBB) induced by CKD in the same hippocampal tissue, double immunofluorescent staining for GFAP and aquaporin-4 (AQP-4) was performed. Brain tissues were incubated overnight at 4 °C in a mixture of mouse anti-GFAP IgG (Millipore; diluted 1:500) and rabbit anti-AQP-4 IgG (Alomone Labs, Jerusalem, Israel; diluted 1:200). After washing with PBS, the sections were incubated in a mixture of Cy2- and Cy3-conjugated secondary antisera (Jackson Immuno Research Labs, West Grove, PA, USA; diluted 1:200) for 3 h at room temperature. Then, the tissues were incubated in DAPI (Invitrogen, Waltham, MA, USA; diluted 1:500) for counterstaining for 15 min at room temperature. After washing with PBS, the slices were placed on a slide and mounted with DPX (Sigma, St. Louis, MO, USA), and then all images were captured using a model Fluoview FV10i and the FV10i software (Olympus, Tokyo, Japan).

### 2.7. Hippocampal Volume in Patients with Cognitive Impairment

This human study was conducted according to the principles expressed in the Declaration of Helsinki. Clinical patient data were obtained from electronic medical records with the approval of the Institutional Review Board, our hospital’s ethics committee (IRB no. 2017-04-014). The Korean version of the Montreal Cognitive Assessment (K-MoCA, English version 7.1, available at www.mocatest.org, accessed on 1 May 2022) was used for assessing cognitive function. We defined mild cognitive impairment as a score of ≤22 in our study [8]. The K-MoCA was scored by a researcher in accordance with the Korean version instructions. Magnetic resonance imaging (MRI) scans were acquired on a 3 Tesla (3T) MRI Philips scanner (Philips Ingenia; Philips Healthcare, Amsterdam, The Netherlands). The detailed method is described in the Appendix A [30,31,32,33,34].

### 2.8. Quantification of Data and Statistical Analysis

Optical fractionation was used to estimate the cell numbers, and the dentate gyrus (DG) and/or CA1 regions were delineated with a 2.5× objective lens for immunodensity quantification. The detailed method is described in the Appendix A. All data obtained from the quantitative measurements were analyzed using one-way analysis of variance (ANOVA) to determine statistical significance. Bonferroni’s test was used for post hoc comparisons. *p*-values of <0.05, 0.01, and 0.001 were considered statistically significant.

## 3. Results

### 3.1. Cognitive Behaviors Deficits in CKD Rat Model

To evaluate whether CKD has an effect on cognition deficit, we performed four cognition-related behavior assays—NOR, Y-maze, Barnes maze, and classical contextual fear conditioning—in the CKD rat models using a previously described procedure. In the NOR test (Figure 1A), the exploration frequency and duration of the naïve and CKD 4-wk rats showed equally adapted exploration patterns between the same two objects. Thus, these were not statistically different (Figure 1(B1,B2)). However, CKD 10-wk rats displayed reduced total exploration times for objects compared with naïve rats (CKD 10 wks, F_1,11_ = 9.36, *p* < 0.01, vs. naïve; F_1,10_ = 10.6, *p* < 0.01, vs. CKD 4 wks; one-way ANOVA; Figure 1(B3)). In the results of the memory phase, the exploration frequency and duration for novel objects in the naïve rats were significantly increased compared with familiar objects (frequency, F_1,14_ = 16.36, *p* < 0.01, duration, F_1,14_ = 42.18, *p* < 0.001; one-way ANOVA; Figure 1(C1,C2)). In CKD 4-wk rats, the exploration time for novel objects was increased (F_1,12_ = 15.82, *p* < 0.01; one-way ANOVA; Figure 1(C2)), whereas the CKD 10-wk rats exhibited similar exploration frequencies and times for recognizing both novel and old objects (Figure 1(C1,C2)). The discrimination index of CKD rats showed a marked reduction compared with that of the naïve rats (CKD 4 wks, F_1,13_ = 11.66, *p* < 0.01, CKD 10 wks, F_1,11_ = 28.96, *p* < 0.001; one-way ANOVA; Figure 1(C3)).

We examined hippocampus-dependent spatial working memory through the Y-maze task with special cues (Figure 2(A1)) and spatial learning and memory through the Barnes maze (Figure 2(B1)). In hippocampus-dependent spatial memory test, the CKD rats showed reduced spontaneous alterations compared with naïve rats (CKD 4 wks, F_1,18_ = 5.46, *p* < 0.05, CKD 10 wks, F_1,16_ = 24.35, *p* < 0.001; one-way ANOVA; Figure 2(A2)). Similarly, to investigate spatial reference memory, the Barnes maze test was performed, and visual cues were placed on each quadrant wall surrounding the platform (Figure 2(B1)). Each animal was trained during an adaptation phase over 4 days, and the time taken for each group to reach the escape hole was measured on the fifth day as a probe day. During the training period, all tested rats showed faster in finding the position of the target hole upon repeated trials (Figure 2(B1)). Notably, CKD 10-wk rats had a higher number of errors for escape hole on the training 2 day and the probe test day (training 2 day, F_1,10_ = 10.43, *p* < 0.01, probe test day, F_1,10_ = 6.23, *p* < 0.05; one-way ANOVA; Figure 2(B2)). In addition, CKD 10-wk rats had longer target latencies on the probe test day (F_1,10_ = 4.39, *p* < 0.05; one-way ANOVA; Figure 2(B3)). These results revealed consistent spatial memory and learning deficits in CKD rats.

Additionally, we investigated whether CKD could cause learning and memory disorders in contextual fear conditioning (Figure 3). In freezing contextual fear learning, CKD 10-wk rats showed decreased freezing times (6 min, F_1,10_ = 6.04, *p* < 0.05, 7 min, F_1,10_ = 7.59, *p* < 0.05; one-way ANOVA; Figure 3A). After 24 h, contextual fear memory deficits were observed in the CKD rats compared with the naïve rats (CKD 4 wks, F_1,10_ = 10.87, *p* < 0.05, CKD 10 wks, F_1,10_ = 18.51, *p* < 0.001; one-way ANOVA; Figure 3B), indicating that CKD led to deficits in hippocampus-dependent memory.

### 3.2. Decreased fEPSP in the Hippocampal Neurons of CKD Rat Model

We additionally investigated activity-dependent synaptic plasticity under the induction of N-methyl-D-aspartate (NMDA) receptor-mediated long-term potentiation (LTP) in CKD rats because the LTP of synaptic activity is an important researched model for learning and memory [34,35]. Figure 4A shows the schema of the in vivo experimental fEPSP recordings from the rat hippocampus. These sTPS stimulations reliably elicited the LTP of the slope of the fEPSP in the CA1 (Figure 4A,B). In CKD rats, the amplitude and the slope of the evoked fEPSP following LTP induction were significantly decreased compared with naïve rats (127.2 ± 17.3% in CKD 4 wks, *p* < 0.01; 109.7 ± 9.6% in CKD 10 wks, *p* < 0.001; 161.1 ± 4.8% in naïve; Figure 4C,D).

### 3.3. Immunoreactivities of Na^+^/H^+^ Exchanger-1 (NHE-1) in CKD Rat Model

To verify acidosis in the hippocampus by CKD uremic toxins, we performed NHE-1 immunostaining. In the naïve rats, NHE-1 immunoreactivity was detected mostly in the hippocampal neurons (Figure 5A–A3). However, NHE-1 expression was enhanced in the neuropil at stratum lacunosum-moleculare (CKD 10 wks, F_1,8_ = 5.79, *p* < 0.05; one-way ANOVA; Figure 5B, arrow; C), molecular layer (CKD 10 wks, F_1,8_ = 12.57, *p* < 0.01; one-way ANOVA; Figure 5B, arrow; D), and hilus (Figure 5B, *) in CKD rats compared with naïve rats. In particular, NHE1 immunoreactivity was reduced in the CA1 region (Figure 5(B2), arrow) and markedly increased in the alveus region of the subiculum and mossy fiber in the CKD 10-wk rats (CKD 10 wks, F_1,8_ = 37.77, *p* < 0.001; one-way ANOVA; Figure 5(B1), *; Figure 5(B3), arrow, E).

### 3.4. Astrogliosis in the Hippocampus of CKD Rat Model

To identify the alterations in the distribution of astroglia expression in the hippocampus following CKD, we performed immunohistochemical analysis for GFAP. In CRF rats, the GFAP immunoreactive astroglia was markedly elevated in the CA1 region and molecular layer of the DG compared with naïve rats (Figure 6(B1–B4)). In brief, the average number of GFAP-positive astrocytes was increased compared with naïve rats (CKD 10 wks, F_1,9_ = 56.4, *p* < 0.001; one-way ANOVA; Figure 6(C1,C3)). Quantitation of the relative densities of GFAP-positive astroglia revealed a result similar to GFAP immunohistochemical expression, which was more than in naïve rats (CKD 10 wks, F_1,9_ = 28.43, *p* < 0.001; one-way ANOVA; Figure 6(C2,C4)). To verify the functional alterations caused by BBB damage in the hippocampus of CKD rats, we performed double immunofluorescent labeling with GFAP and AQP-4 (Figure 6D–G). In CKD rats, AQP4-positive immunoreactivity in the astrocytes and vessels was significantly increased in the hippocampal CA1 and DG regions compared with naïve rats (CKD 10 wks, CA1, F_1,14_ = 107.55, *p* < 0.001; DG, F_1,12_ = 107.55, *p* < 0.001; one-way ANOVA; Figure 6E,G,(H1),(H2)).

### 3.5. Hippocampal Volume in CKD Patients with Cognitive Impairment

There was no difference in age between the controls (*n* = 34) and end-stage renal disease (ESRD) hemodialysis patients (*n* = 42). There was also no difference in age between ESRD patients with cognitive impairment (*n* = 17) and without cognitive impairment (*n* = 25). Relative to the controls, the ESRD patients showed a significant decrease in hippocampal volume (Figure 7). Moreover, the ESRD patients with cognitive impairment showed a significant decrease in hippocampal volume compared with ESRD patients without cognitive impairment. 

## 4. Discussion

The present study showed that CKD was associated with cognition-related behavioral deterioration, BBB disruption, and a decrease in synaptic plasticity. In addition, we showed a decrease in hippocampal volume in ESRD patients with cognitive disorders compared with ESRD without cognitive impairment or the controls. These findings provide evidence suggesting an association between cognitive impairment and vascular disorders in CKD patients. The available behavior studies showed similar disturbances involving cognitive dysfunction in the 5/6 nephrectomy model [36,37], although there were no expected behavior changes in the CKD rat model [38,39].

Recently, Renczés et al. conducted behavior tests (open field, light/dark transition, and novel object recognition tests) at 2, 4, and 6 months after 5/6 nephrectomy [39]. The CKD rats showed shorter distances and spent less time in the center zone at 2 and 4 months. However, the open-field ambulation returned back to baseline level at 6 months in CKD rats. They did not show significant differences between the CKD and control rats in any of the observed behaviors. Discrepancies from our results may arise from variability in the susceptibility to cognitive impairment in CKD animals, progression of CKD after 5/6 nephrectomy, and repeated behavior testing.

To assess hippocampal-dependent memory, we used the novel object recognition test, Y-maze with a special cue, Barnes maze, and classical contextual fear conditioning test. Memory deficiencies were verified using the novel object recognition test and spatial working memory test [40]. Several studies have reported on cognitive dysfunction in animal models of CKD [36,37,38,41]. In our study, CKD rats exhibited decreased novel object recognition memory, consistent with previous reports. In addition, we verified impaired spatial working memory and deficits in 24-h contextual fear memory. The hippocampus is part of a system of structures in the medial temporal lobe that is essential for memory, as they play roles in the acquisition and retrievability of spatial knowledge [42]. Contextual fear memory is an associative memory dependent upon the hippocampus [43]. According to previous studies, enhanced IL-1β expression is disturbed processes of contextual fear conditioning, and this response is affected by uremic toxin due to chronic kidney disease [44,45,46]. However, studies on the correlation between uremic toxin and inflammatory reactivity according to chronic kidney disease are still insufficient, and further studies are needed. In addition, the expression of contextual fear memory showed differences according to gender in the previous study results [47]. Therefore, mechanistic differences may occur for hippocampal dysfunction caused by chronic renal failure according to gender, and various memory-related behavioral tests and various effect mechanism studies are needed to investigate the relationship. Our results revealed that CKD was involved in poor hippocampal-dependent short-term spatial working memory and spatial learning and memory. Our findings showed a decline in hippocampal-dependent memory according to CKD progression, and this suggests that hippocampal dysfunction may have an important role in cognitive impairment in CKD patients.

Memory acquisition is based on changes in synaptic efficiency that permit strengthening of the associations between neurons, and activity-dependent synaptic plasticity at appropriate synapses during memory formation is essential for the storage of information [48]. LTP is a complex response dependent upon changes in ionic distribution (especially of Ca^2+^) and the activation of a number of different intracellular biochemical mechanisms [49]. Indeed, the correlation between enhanced LTP in the CA1 and better learning performance has been reported in previous studies [50,51,52]. The present findings suggest that prolonged, reduced LTP induction after CKD may be accompanied by long-term memory deficits due to impairments in synaptic plasticity. Moreover, improving hippocampal LTP may be a direct mechanism for improving learning and memory behavior in CKD rats [53,54,55].

NHE-1 is a membranous protein and ubiquitously expressed when neurons adjust their intracellular pH to mediate DNA synthesis, cell volume, neurotransmission, and protein function and degradation in the initiation of cellular growth and differentiation [56,57]. The activation of NHE regulates extra- and intracellular pH and affects intracellular Na^+^ concentration, which, in turn, can alter the function of a number of ion channels and result in changes in neuronal excitability [58]. Among the NHE family, NHE-1 plays an important role in regulating physiological and pathophysiological processes in brain diseases [59,60,61]. NHE-1 activity is increased by intracellular acidosis caused by the interaction of intracellular H^+^ in the myocardium and renal tissues [62,63,64]. In addition, the hippocampal neurons and astrocytes upregulate NHE-1 expression by intracellular acidosis, and NHE1 plays an essential role in acid extrusion after acidosis in cortical neurons [65,66,67]. However, there are no published studies on changes in hippocampal NHE-1 expression in 5/6 nephrectomy rats. To extend our knowledge, we reported that NHE-1 expression was enhanced in the hippocampus of CKD rats. A previous study showed that a broad-spectrum NHE inhibitor improved LTP induced by a theta burst in the hippocampal slices in both young and senescent animals [68]. Thus, we think that changes in the NHE-1 expression may affect intracellular pH, which could affect cognitive function. In addition, alkali therapy may attenuate the progression of kidney injury and ameliorate accompanying cognitive deficiency by inhibiting NHE in 5/6 nephrectomy rats [69].

Aquaporin channels regulate water transport for maintaining water homeostasis, and its functional abnormalities are implicated in several disease pathways [70]. Glial AQP-4 regulates extracellular matrix volume, potassium buffering, cerebrospinal fluid circulation, neuroinflammation, osmosensation, and calcium signaling [71]. A previous study reported that mice lacking AQP-4 were partially protected from brain edema in water intoxication and ischemic brain injury [72]. As astrocyte markers, AQP-4 and GFAP have been related to several physiological and pathological conditions in the central nervous system as well as in BBB breakdown [73]. Some studies suggest a causal relationship between BBB disruption and cognitive dysfunction in aging, diabetes, and Alzheimer’s disease [74,75,76]. Recently, indoxyl sulfate was shown to mediate BBB disruption, leading to cognitive impairment during experimental renal dysfunction [77]. Uric acid passed through the BBB and resulted in the accumulation of gliosis in the hippocampus [78]. Furthermore, CKD affects several organs, including the nervous system, due to the retention of uremic toxins, electrolytes, and water, causing metabolic disturbances [79,80]. In the present study, we observed that GFAP expression was increased, and the coexpression with AQP-4 in the hippocampal CA1 and DG regions was enhanced. Therefore, our results may indicate that CKD disrupts the BBB.

The hippocampus has a role in working memory, processing speed, and executive functioning, in addition to encoding and storage [81]. A decline in hippocampal volume has been well observed in aging, epilepsy, ethanol ingestion, and Alzheimer’s disease [14,82,83,84]. We found that the volume of the hippocampus was smaller in hemodialysis patients than in the general population, and it was also shown to be smaller in hemodialysis patients with cognitive impairment. Various causes have been associated with cognitive dysfunction in CKD. Thus, our clinical results also suggest that the hippocampus is an area vulnerable to the effects of CKD. A limitation of the current study is that the relationship between the level of uremia in the hippocampus according to the level of chronic renal failure according to the GFR has not been clearly identified. In future studies, it is thought that pharmacological treatment effects can be applied to clinical patients more efficiently if comparative studies are conducted to prove the effect of antagonists and if GFR is measured.

## 5. Conclusions

The present study suggested uremia due to kidney damage may cause destruction to the BBB by acidosis in the brain and, thus, may lead to recognition dysfunction in hippocampal association following CKD. However, this hypothesis requires definitive knowledge of the pathophysiology mechanisms.

## Figures and Tables

**Figure 1 brainsci-12-00905-f001:**
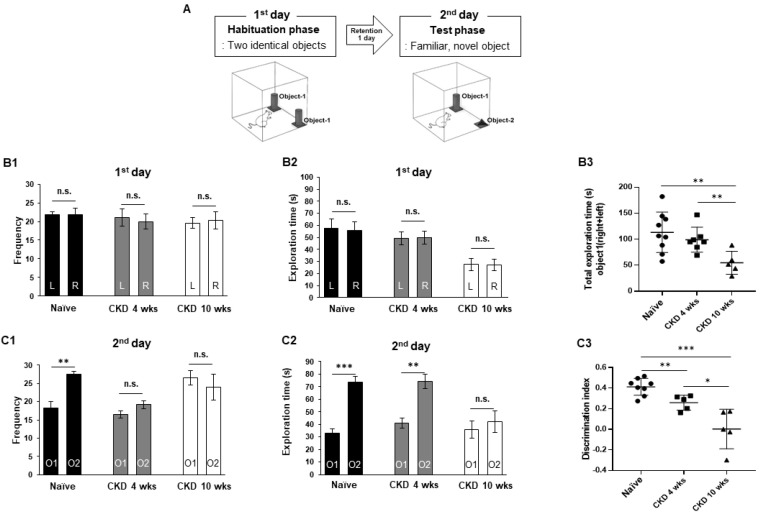
Decreased novel object recognition memory in CKD model rats. Schema of the novel object recognition test (**A**). In the habituation period, the frequency of searching for familiar and novel objects was similar in naïve and CKD 4, 10- wk rats (**B1**). The exploration time for familiar and novel objects was similar in both naïve and CKD rats (**B2**). However, the object exploration times were markedly decreased in CKD 10-wk rats compared with naïve rats (**B3**). Naïve rats showed significantly increased frequency and times for novel objects (**C1**,**C2**). CKD 4-wk rats showed novelty preferences in exploration times (**C2**). However, CKD 10-wk rats did not show preferences for novel objects (**C1**,**C2**). In particular, the discrimination index of CKD rats for new objects was notably different as time passed after surgery (**C3**). L, left side; R, right side; O1, object-1; O2, object-2. Data are presented as means ± standard errors of the mean. n.s.: not significant, * *p* < 0.05, ** *p* < 0.01, *** *p* < 0.001 by one-way analysis of variance naïve: *n* = 8; CKD 4 wks: *n* = 7; CKD 10 wks: *n* = 5.

**Figure 2 brainsci-12-00905-f002:**
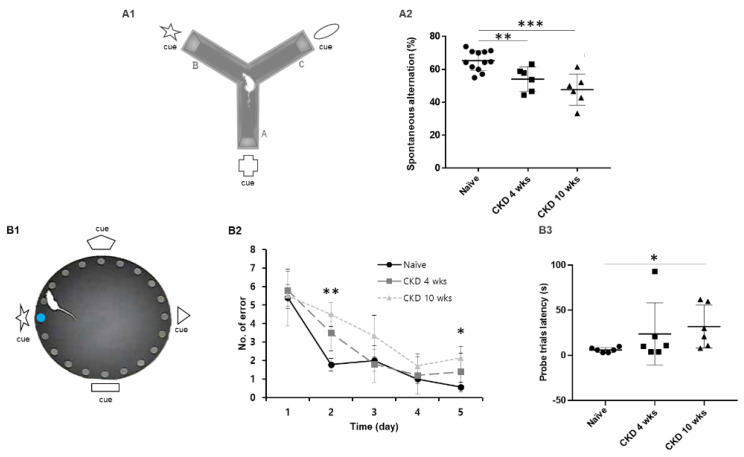
Decreased spatial working memory in CKD model rats. A schematic diagram showing the special cues working memory test (**A1**). In the CKD rats, the average percentage of spontaneous movement was decreased compared with naïve rats (**A2**). Barnes maze apparatus was used to examine spatial learning and memory in CKD model rats (**B1**). The number of errors for escape hole was reduced for all rats upon repeated trials (**B2**). CKD 10-wk rats had a higher time of error on the second day and fifth day (probe test) compared with naïve rats (**B2**). The latent time to find the target hole increased on the probe test day in the CKD 10 wks compared with naïve rats. (**B3**). The blue round symbol in B1indicates the escape hole. Data are presented as means ± SEM. * *p* < 0.05, ** *p* < 0.01, *** *p* < 0.001 by one-way analysis of variance. Y-maze, naïve: *n* = 12; CKD 4 wks: *n* = 7; CKD 10 wks: *n* = 5; Barnes maze: *n* = 6/group.

**Figure 3 brainsci-12-00905-f003:**
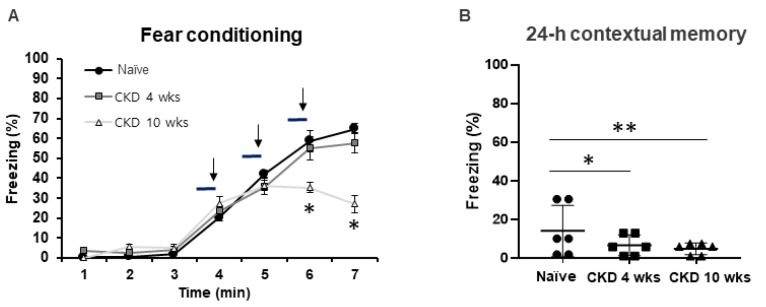
Impairment of contextual fear memory in CKD 4, 10- wk rats. Contextual and cued fear conditioning and 24-h retrieval of contextual memory were performed (**A**,**B**) No differences in classical fear conditioning were observed in CKD 4-wk rats, although CKD 10-wk rats spent a significantly lower percentage of time freezing compared with naïve rats (**A**). The horizontal line indicates the duration of a tone (28 s), and the vertical arrow indicates the time of foot shocks (2 s). Deficits in contextual fear memory were observed in the CKD rats (**B**). Data are presented as means ± SEM. * *p* < 0.05, ** *p* < 0.01 by one-way analysis of variance, contextual fear conditioning: *n* = 6/group.

**Figure 4 brainsci-12-00905-f004:**
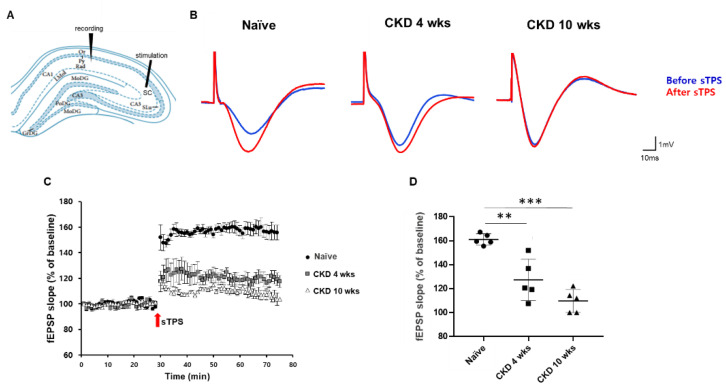
Decreased fEPSP in the hippocampal CA1 regions of CKD model rats. A schematic diagram showing the in vivo experimental setup for recording fEPSP in the rat hippocampus (**A**). Representative traces for fEPSP in the hippocampal CA1 region before and after single-pulse stimuli in naïve and CKD rats (**B**). The amplitude of evoked fEPSP (**C**) and the slope changes (**D**) following LTP in CKD rats were attenuated compared with naïve rats. The arrows in panel B indicate the time point of strong theta-patterned stimulus (sTPS) application for LTP. Data are presented as means ± SEM. ** *p* < 0.01, *** *p* < 0.001 by one-way analysis of variance, fEPSP: *n* = 5/group.

**Figure 5 brainsci-12-00905-f005:**
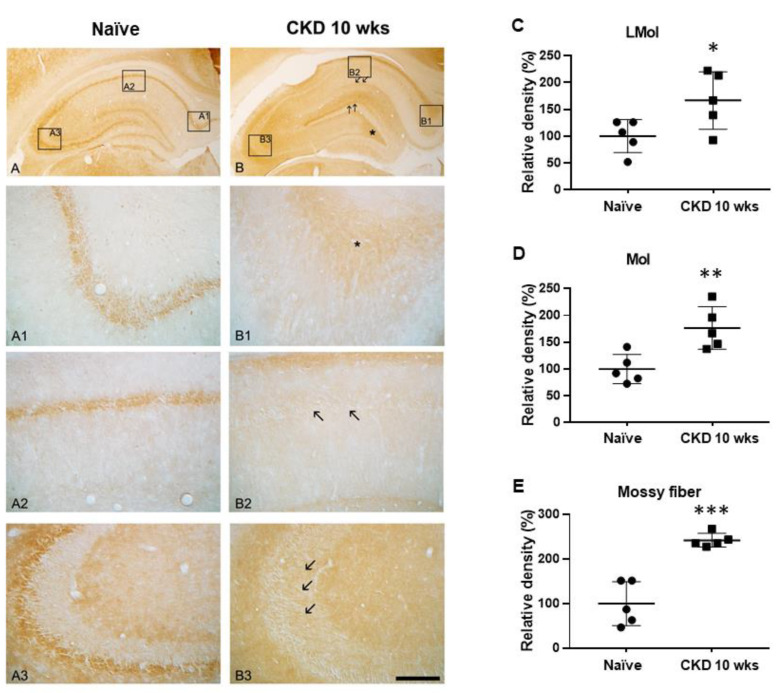
Upregulation of NHE1 immunoreactivity in the hippocampus of CKD model rats. NHE1 immunoreactivity in the hippocampus of naïve rats was mainly observed in the neurons (**A**): subiculum (**A1**), CA1 (**A2**), and CA2-3 (**A3**). CKD did not affect NHE1 expression in the neurons of the hippocampus ((**B**) and arrows of (**B2**)). NHE1 expression was enhanced in the neuropil at stratum lacunosum-moleculare, molecular layer (arrows), and hilus (*) regions in the CKD rats (**B**). CKD rats showed increased NHE1 immunoreactivity in the subiculum and alveus regions ((**B1**), *) and mossy fiber ((**B3**), arrows) of the hippocampus compared with naïve rats. NHE1 expression in the CA1 region was decreased in the neurons of CKD rats ((**B2**), arrows). The NHE1 densitometric results were similar to the immunohistochemical data ((**C**), LMol; (**D**), Mol; (**E**), CA2-3). Scale bar = 400 μm (panels (**A**,**B**)), 50μm (panels (**A1**–**A3**) and (**B1**–**B3**)). * *p* < 0.05, ** *p* < 0.01, *** *p* < 0.001 by one-way analysis of variance, *n* = 5/group.

**Figure 6 brainsci-12-00905-f006:**
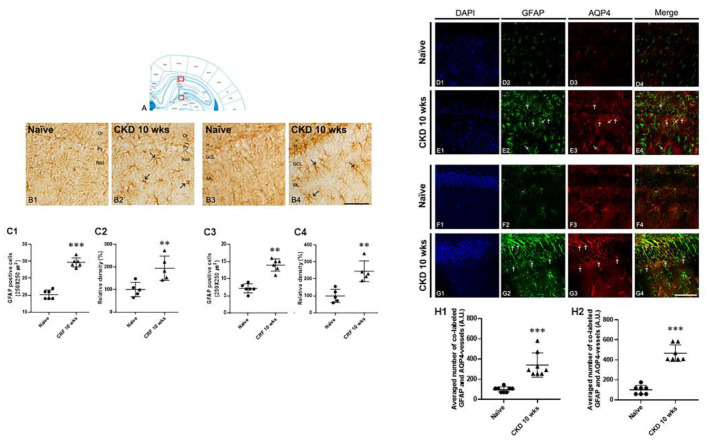
Gliosis in the hippocampal region of CKD model rats. Diagram of the rat hippocampus (**A**). GFAP immunoreactivity in the hippocampus of CKD 10-wk rats was enhanced in the CA1 and DG regions (arrows) compared with naïve rats (**B1**–**B4**): scale bar = 25 μm. The average number of GFAP-positive astroglia and the densitometric analysis results of GFAP immunoreactivity were consistent with the immunohistochemical data (**C1**–**C4**). Double labeling of GFAP and AQP-4 in the hippocampal CA1 and DG regions of naïve rats (**D**,**F**) and CKD 10-wk rats (**E**,**G**): GFAP (green); AQP-4 (red); merged images (yellow); scale bar = 18.8 μm; A.U., arbitrary unit. Data are presented as the mean and SEM. Significant differences from naïve rats (**H1**,**H2**), ** *p* < 0.01, *** *p* < 0.001 by one-way analysis of variance, IHC: *n* = 5/group; CA1 of double labeling: *n* = 8/group; DG of double labeling: *n* = 7/group.

**Figure 7 brainsci-12-00905-f007:**
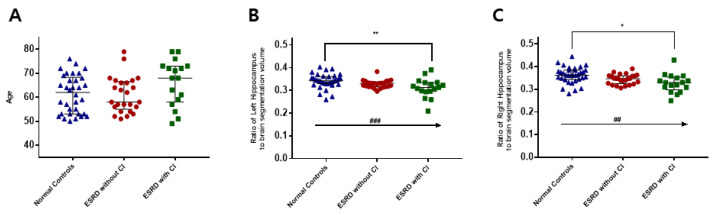
Hippocampal volume in control and end-stage renal disease patients with/without cognitive impairment. There was no difference among the groups (**A**). The ESRD patients with or without cognitive impairment had smaller left and right hippocampal volumes than the control subjects (**B**,**C**). ESRD patients with cognitive impairment had smaller left and right hippocampal volumes than ESRD patients without cognitive impairment (**B**,**C**). * *p* < 0.05, ** *p* < 0.01.

## Data Availability

Not applicable.

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
