# Peer review of "Cognitive Sequelae and Hippocampal Dysfunction in Chronic Kidney Disease following 5/6 Nephrectomy"

_brainsci, 2022, doi:10.3390/brainsci12070905_

Round 1

Reviewer 1 Report

In the present manuscript Dr Yu and colleagues wanted to verify whether chronic kidney desease, induced by a serimental model, may induced cognitive impairment and hippocampal volume loss, paralleled by deficits in synaptic plasticity and behavior depending on impairments in the hippocampal formation. They observed dysfunction and intrinsic synaptic properties in the hippocampus paralleled by deficits in novel object recognition (NOR) memory, spatial working memory, 24-hr contextual fear memory, and hippocampal synaptic plasticity. All these impairments are related to uremia, demostrating a well known brain-kidney crosstalk that, in this case, caused hippocampus-relevant cognitive impairment. Overall the manuscript is potentially interesting but I think that there are some issues that need to be addressed before considering it suitable for publication.

1.      All experiments utilized Sprague-Dawley (SD) rats (8 weeks old) but it is not clear if they used male or female individuals. It seems evident from what they wrote in the abstract but if they used males only please specify why females are leaved out of the study.

2.      It is not clear if during nephrectomy, where the upper and lower poles of the left kidney were resected using surgical scissors and then, a unilateral nephrectomy was performed at the right site, they elimitnate also the adrenal glands? At wich age the surgical procedure was made?

3.      The results obtained in the Barnes maze are a little odd to me! In fact from NOR test it result that the exploration time between naive and CKD 4 weeks is not different suggesting that the motility of the animals seems not altered by the treatment right? This may appear from graph B2 of Fig. 1 where the exploration time is significantly lower in 10weeks after CKD when compared with NAIVE. Anyway, surprisingly the 1st day (habituation) of the Barnes maze, where all animals see for the first time the arena, they probably explore in the same way the arena without any significant difference between group unless some problem in the motility may occur caused by the treatment. The latency in CKD animals are strongly impaired during the 1st day compared with controls (fig 2B2). Did the author explore the motility performance in all groups? Why in the NAIVE group there is no improvement in the latency in finding the escape chamber during training days? This is weird! Paradoxically it seems that treated animals “learn better” than controls since they have a greater reduction in latency during the training period when compared with controls!! Please report also the wrong pokes before finding the right hole (escape chamber). 

4.      Authors affirm for electrophysiological recordings “A stable baseline was recorded for 30 – 60 min ………. expressed as percentages of the mean fEPSP slope measured during the 30-min baseline period”. This is not true by scatter plots in Fig. 4C since only 10 min of baseline were reported. Please put in graph the whole period of control and not only 10 min of it.

5.      Behavioral data are quite no discussed in the discussion section. The author wrote “In our study, CKD rats exhibited decreased novel object recognition memory, consistent with previous reports. In addition, we verified impaired spatial working memory and deficits in 24-hr contextual fear memory. The hippocampus is part of a system of structures in the medial temporal lobe, which are essential for memory as they play roles in the acquisition and retrievability of spatial knowledge [35]. Contextual fear memory is associative memory dependent upon the hippocampus [36]. Our results revealed that CKD was involved in poor hippocampal- dependent short-term spatial working memory and spatial learning and memory”…..This is not a discussion but only what they found! This part need absolutely to be widely addressed.

6.      The authors conclude “The present study suggested uremia due to kidney damage may cause destruction to 436 the BBB by acidosis in the brain and thus, may lead to recognition dysfunction in hippo- 437 campal association following CKD”…..But to me it’s hard to conclude that all the observed effects are due to uremia. Did they measure the level of uremia in the present study? Did they tried to antagonize this effect? I understand that other authors observed the effect of uremia but, in the present work, how can they be sure that uremia is really involved in the observed effects?

Author Response

Reviewer 1

In the present manuscript Dr Yu and colleagues wanted to verify whether chronic kidney disease, induced by a serimental model, may induced cognitive impairment and hippocampal volume loss, paralleled by deficits in synaptic plasticity and behavior depending on impairments in the hippocampal formation. They observed dysfunction and intrinsic synaptic properties in the hippocampus paralleled by deficits in novel object recognition (NOR) memory, spatial working memory, 24-hr contextual fear memory, and hippocampal synaptic plasticity. All these impairments are related to uremia, demostrating a well known brain-kidney crosstalk that, in this case, caused hippocampus-relevant cognitive impairment. Overall the manuscript is potentially interesting but I think that there are some issues that need to be addressed before considering it suitable for publication.

Comment 1.      All experiments utilized Sprague-Dawley (SD) rats (8 weeks old) but it is not clear if they used male or female individuals. It seems evident from what they wrote in the abstract but if they used males only please specify why females are leaved out of the study.

Response: In order to perform rodent ‘behavioral study’, we used ‘only male rats’. The aim of our current study is to determine whether CKD contributes to cognitive deficits in male rats, and we found clear evidences from our behavioral studies. As reviewer mentioned, if we perform behavioral test using female CKD rats in future study, we will be able to identify sexual differences in CKD-induced cognitive behaviors

Comment 2.      It is not clear if during nephrectomy, where the upper and lower poles of the left kidney were resected using surgical scissors and then, a unilateral nephrectomy was performed at the right site, they elimitnate also the adrenal glands? At which age the surgical procedure was made?

Response: We protected adrenal glands and to nephrectomy, we eliminate a peripheral fat pad of kidney and then proceed 5/6 nephrectomy at the 8 weeks old male rats. We also re-write experimental animal part of manuscript with these detail surgical methods (Correction version, Page 2 Line 86).

Comment 3.      The results obtained in the Barnes maze are a little odd to me! In fact from NOR test it result that the exploration time between naive and CKD 4 weeks is not different suggesting that the motility of the animals seems not altered by the treatment right? This may appear from graph B2 of Fig. 1 where the exploration time is significantly lower in 10weeks after CKD when compared with NAIVE. Anyway, surprisingly the 1st day (habituation) of the Barnes maze, where all animals see for the first time the arena, they probably explore in the same way the arena without any significant difference between group unless some problem in the motility may occur caused by the treatment. The latency in CKD animals are strongly impaired during the 1st day compared with controls (fig 2B2). Did the author explore the motility performance in all groups? Why in the NAIVE group there is no improvement in the latency in finding the escape chamber during training days? This is weird! Paradoxically it seems that treated animals “learn better” than controls since they have a greater reduction in latency during the training period when compared with controls!! Please report also the wrong pokes before finding the right hole (escape chamber). 

Response: First, the Fig-1B2 graph shows “no spatial preference for left or right side”, because objects in left and light side are same (object 1 is placed on both sides). The mentioned motility is far from what we want to present through Fig1B2. For information on motility for CKD, please refer to our previous paper (manuscript Rf. 17) attached below.

Yu, Y.H.; Kim, S.W.; Park, D.K.; Song, H.Y.; Kim, D.S.; Gil, H.W. Altered Emotional Phenotypes in Chronic Kidney Disease Following 5/6 Nephrectomy. Brain Sci 2021, 11, doi:10.3390/brainsci11070882.

Second, following reviewer’s suggestion, we additionally graph for number of finding wrong pokes to clarify differences in learning and memory between groups (Correction version, Fig 2B2).

Comment 4.      Authors affirm for electrophysiological recordings “A stable baseline was recorded for 30 – 60 min expressed as percentages of the mean fEPSP slope measured during the 30-min baseline period”. This is not true by scatter plots in Fig. 4C since only 10 min of baseline were reported. Please put in graph the whole period of control and not only 10 min of it.

Response:  Following the reviewer’s comments, we replaced the Fig. 4C. with new fEPSP slope showing whole period including 30 min of baseline.

Comment 5.      Behavioral data are quite no discussed in the discussion section. The author wrote “In our study, CKD rats exhibited decreased novel object recognition memory, consistent with previous reports. In addition, we verified impaired spatial working memory and deficits in 24-hr contextual fear memory. The hippocampus is part of a system of structures in the medial temporal lobe, which are essential for memory as they play roles in the acquisition and retrievability of spatial knowledge [35]. Contextual fear memory is associative memory dependent upon the hippocampus [36]. Our results revealed that CKD was involved in poor hippocampal- dependent short-term spatial working memory and spatial learning and memory”…..This is not a discussion but only what they found! This part need absolutely to be widely addressed.

 Response: Following reviewer’s suggestion, we added more information about contextual fear memory on CKD in discussion part of manuscript as below (Correction version, Page 10 Line 382):

According to previous studies, enhanced IL-1β expression is disturbed processes of contextual fear conditioning and this response is affected by uremic toxin due to chronic kidney disease (ref 1-3). However, studies on the correlation between uremic toxin and inflammatory reactivity according to chronic kidney disease are still insufficient and further studies are needed. In addition, the expression of contextual fear memory showed differences according to gender in the previous study results (ref 4). Therefore, mechanistic differences may occur for hippocampal dysfunction caused by chronic renal failure according to gender, and various memory-related behavioral tests and various effect mechanism study are needed to study the relationship.

  1. Palin K, Bluthé RM, Verrier D, Tridon V, Dantzer R, Lestage J. Interleukin-1beta mediates the memory impairment associated with a delayed type hypersensitivity response to bacillus Calmette-Guérin in the rat hippocampus. Brain Behav Immun. 18:223–230 (2004)
  2. Franco A de O, Starosta RT, Roriz-Cruz M, The specific impact of uremic toxins upon cognitive domains: a review, J Bras Nefrol. 41(1): 103–111 (2019)
  3. Rossi M, Canpbell KL, Johnson DW et al., Protein-bound uremic toxins, inflammation and oxidative stress: a cross-sectional study in stage 3-4 chronic kidney disease, Arch Med Res. 45(4):309-17 (2014)
  4. Russo, A.S. and Parsons, R.G. Behavioral Expression of Contextual Fear in Male and Female Rats. Front Behav Neurosci. 2021, 15:671017, doi: 10.3389/fnbeh.2021.671017. eCollection 2021.

.

Comment 6.      The authors conclude “The present study suggested uremia due to kidney damage may cause destruction to the BBB by acidosis in the brain and thus, may lead to recognition dysfunction in hippocampal association following CKD”…..But to me it’s hard to conclude that all the observed effects are due to uremia. Did they measure the level of uremia in the present study? Did they tried to antagonize this effect? I understand that other authors observed the effect of uremia but, in the present work, how can they be sure that uremia is really involved in the observed effects?

Response: Our previous study was established a chronic renal failure model by demonstrating through creatinine levels and histological changes of kidney staining for the chronic renal failure model. We would be grateful if you could refer to the results of previous research (manuscript Rf. 17). CKD patients were diagnosed by the decline of kidney function through the measurement of serum creatinine. Therefore, it may that these changes in creatinine by CKD could induce uremia. In fact, in the present study, we are confirmed to enhance expression of NHE-1 in the CKD hippocampus, these results suggest pH imbalance of the brain. In addition, gliosis of astrocytes around blood vessels in the hippocampus shows changes in the BBB caused by acidosis following CKD. Future research about present results, we plan to confirm the therapeutic effect of cognitive dysfunction caused about uremia by measuring the pH and proving the effect on antagonist drugs.

  1. Book: Uremia, StatPearls Publishing; 2022, Michael R. Zemaitis; Lisa A. Foris; Shravan Katta; Khalid Bashir.

Reviewer 2 Report

I have had a pleasure of reviewing the following manuscript “Cognitive sequelae and hippocampal dysfunction in chronic kidney disease following 5/6 nephrectomy

In this study, authors presented evidence on a long standing question whether CKD affects cognitive deficits. If so, what could be the key cause of it?

To answer these questions, authors utilized CKD rat model and conducted various related experiments to evaluate cognitive performance in rats.

Authors showed that CKD 10 weeks (wk) rats showed reduced total exploration time for objects compared to control rats. The discrimination index of CKD rats showed a marked reduction compared to that of the control rats. Secondly, in hippocampus-dependent spatial-memory test, the CKD rats showed reduced spontaneous alterations compared to control rats. As evaluated by Branes maze test, CKD 10 wk rats had longer target latencies on training day 1 and probe test day suggesting compromised spatial memory in CKD rats. In freezing contextual fear learning, CKD 10 wk rats showed decreased freezing time suggesting defective contextual fear memory. Next they showed that the synaptic plasticity is decreased in hippocampal neurons of CKD rats.

CKD associated uremia causes acidosis. They suspected if the acidosis plays a role in a poor hippocampal structure and function? They showed NHE1 immunoreactivity was increased in the hippocampus of CKD rats. They further showed the evidence for astrogliosis in the hippocampus of CKS rats. Ultimately, authors evaluated the hippocampal volume in CKD patients with cognitive impairments.

The present study showed that CKD affects cognition, blood brain barrier, decreases synaptic plasticity. Furthermore, they showed decrease in hippocampal volume in ESRD patients with cognitive deficits compared with ESRD without cognitive issues or the controls. These findings provide evidence suggesting an association between cognitive impairment and vascular disorders in CKD patients.

Line 58-62 could be rephrased and broken into two separate sentence to make a clear point.

This study was well-designed and the paper is well-written. I personally like the discussion part where authors interpreted the study well.

Author Response

I have had a pleasure of reviewing the following manuscript “Cognitive sequelae and hippocampal dysfunction in chronic kidney disease following 5/6 nephrectomy

In this study, authors presented evidence on a long standing question whether CKD affects cognitive deficits. If so, what could be the key cause of it?

To answer these questions, authors utilized CKD rat model and conducted various related experiments to evaluate cognitive performance in rats.

Authors showed that CKD 10 weeks (wk) rats showed reduced total exploration time for objects compared to control rats. The discrimination index of CKD rats showed a marked reduction compared to that of the control rats. Secondly, in hippocampus-dependent spatial-memory test, the CKD rats showed reduced spontaneous alterations compared to control rats. As evaluated by Branes maze test, CKD 10 wk rats had longer target latencies on training day 1 and probe test day suggesting compromised spatial memory in CKD rats. In freezing contextual fear learning, CKD 10 wk rats showed decreased freezing time suggesting defective contextual fear memory. Next they showed that the synaptic plasticity is decreased in hippocampal neurons of CKD rats.

CKD associated uremia causes acidosis. They suspected if the acidosis plays a role in a poor hippocampal structure and function? They showed NHE1 immunoreactivity was increased in the hippocampus of CKD rats. They further showed the evidence for astrogliosis in the hippocampus of CKS rats. Ultimately, authors evaluated the hippocampal volume in CKD patients with cognitive impairments.

The present study showed that CKD affects cognition, blood brain barrier, decreases synaptic plasticity. Furthermore, they showed decrease in hippocampal volume in ESRD patients with cognitive deficits compared with ESRD without cognitive issues or the controls. These findings provide evidence suggesting an association between cognitive impairment and vascular disorders in CKD patients.

Line 58-62 could be rephrased and broken into two separate sentence to make a clear point.

This study was well-designed and the paper is well-written. I personally like the discussion part where authors interpreted the study well.

Response: We are very grateful for reviewer #2’s critical and encouraging comments. In addition, following reviewer’s comments, we have rephrased Line 58-62, and separated into two sentences as below.

Line 58-62: “In the present study, to verify whether CKD induced cognitive impairment and hippocampal volume loss, and deficits of hippocampal synaptic functions, we established a stable rat model of uremic CKD by performing 5/6 nephrectomies and measured cognitive dysfunction and intrinsic synaptic properties in the hippocampus by performing behavioral and electrophysiological tests.”

Above sentence has been rephrased as follow (Correction version, Page 2 Line 66). “Here, in order to verify whether CKD contributes to cognitive deficits, we firstly performed 5/6 nephrectomy (5/6 Nx) and established a stable rat model of CKD. We next measured cognitive dysfunction and intrinsic synaptic properties in the hippocampus by performing behavioral, anatomical, and electrophysiological tests in CKD rat model.”

Reviewer 3 Report

The issue concerning the impact of comorbidities related to internal medicine impacting cognitive functioning. In this study authors elaborate on the role of dysfunction of chronic kidney disease. I have several point which should be additionally discussed:

1. In the introduction and discussion authors should also elaborate on comorbidities, which could additionally impact or be interpreted as a risk factor of chronic kidney disease as glycemic variability and hypertension. Glycemic variability was also recently described as a risk factor of hippocampal hypoperfusion -

Ref.

The Rate of Decrease in Brain Perfusion in Progressive Supranuclear Palsy and Corticobasal Syndrome May Be Impacted by Glycemic Variability-A Pilot Study. Front Neurol. 2021 Nov 8;12:767480. doi: 10.3389/fneur.2021.767480. PMID: 34819913; PMCID: PMC8606811.

2. A separate paragraph indicating methodological limitations of the study should be added. 

3. Authors should present their point of view on future perspectives.

Author Response

Reviewer 3

The issue concerning the impact of comorbidities related to internal medicine impacting cognitive functioning. In this study authors elaborate on the role of dysfunction of chronic kidney disease. I have several point which should be additionally discussed:

Comment 1. In the introduction and discussion authors should also elaborate on comorbidities, which could additionally impact or be interpreted as a risk factor of chronic kidney disease as glycemic variability and hypertension. Glycemic variability was also recently described as a risk factor of hippocampal hypoperfusion –

Response: Thank you for good comment. I agree that comorbidities clinically impact the cognitive function. Following reviewer’s suggestion, we added more explanation in introduction part of manuscript as below (Correction version, Page 2 Line 47).

Many comorbidities including hypertension, diabetes mellitus, and CKD are risk factors for cognitive impairment. Recent study showed glycemic variability could influence on brain perfusion, which decrease in perfusion in the hippocampus are may impact the rate of cognitive deterioration (6)

  1. The Rate of Decrease in Brain Perfusion in Progressive Supranuclear Palsy and Corticobasal Syndrome May Be Impacted by Glycemic Variability-A Pilot Study. Front Neurol. 2021 Nov 8;12:767480. doi: 10.3389/fneur.2021.767480. PMID: 34819913; PMCID: PMC8606811.

Comment 2. A separate paragraph indicating methodological limitations of the study should be added. 

With respect to reviewer’s comments, we re-write about limitation of our current study in the conclusion part (Correction version, Page 11 Line 456)

  1. Authors should present their point of view on future perspectives.

Following the reviewer’s comments, we re-write about future perspective of our current study in the conclusion part (Correction version, Page 11 Line 456)

Round 2

Reviewer 1 Report

I'm satisfied about the authors answers. The manuscript is now enproved

Reviewer 3 Report

Authors have implemented changes and I do not have further comments.